# Factors associated with nonadherence to the American Academy of Pediatrics 2014 bronchiolitis guidelines: A retrospective study

**Laure F. Pittet** [1,2]*, **Alban Glangetas**[3], **Constance Barazzone-Argiroffo**[4], **Alain Gervaix**[3], **Klara M. Posfay-Barbe**[1,2], **Annick Galetto-Lacour**[3], **Fabiola Stollar**[1]

**1** Division of General Pediatrics, Department of Pediatrics, Gynecology & Obstetrics, University Hospitals of Geneva and University of Geneva's Faculty of Medicine, Geneva, Switzerland, **2** Unit of Pediatric Infectious Diseases, Department of Pediatrics, Gynecology & Obstetrics, University Hospitals of Geneva and University of Geneva's Faculty of Medicine, Geneva, Switzerland, **3** Division of Pediatric Emergency, Department of Pediatrics, Gynecology & Obstetrics, University Hospitals of Geneva and University of Geneva's Faculty of Medicine, Geneva, Switzerland, **4** Unit of Pediatric Pulmonology, Department of Pediatrics, Gynecology & Obstetrics, University Hospitals of Geneva and University of Geneva's Faculty of Medicine, Geneva, Switzerland

* laure.pittet@hcuge.ch

**Data Availability Statement:** The data underlying the results presented in the study are available

## Abstract

The latest guideline from the American Academy of Pediatrics for the management of bronchiolitis has helped reduce unnecessary interventions and costs. However, data on patients still receiving interventions are missing. In patients with acute bronchiolitis whose management was assessed and compared with current achievable benchmarks of care, we aimed to identify factors associated with nonadherence to guideline recommendations. In this single-centre retrospective study the management of bronchiolitis pre-guideline (Period 1: 2010 to 2012) was compared with two periods post-guideline (Period 2: 2015 to 2016, early post-guideline; and Period 3: 2017 to 2018, late post-guideline) in otherwise healthy infants aged less than 1 year presenting at the Children's University Hospitals of Geneva (Switzerland). Post-guideline, bronchodilators were more frequently administered to older (>6 months; OR 25.8, 95%CI 12.6–52.6), and atopic (OR 3.5, 95%CI 1.5–7.5) children with wheezing (OR 5.4, 95%CI 3.3–8.7). Oral corticosteroids were prescribed more frequently to older (>6 months; OR 5.2, 95%CI 1.4–18.7) infants with wheezing (OR 4.9, 95% CI 1.3–17.8). Antibiotics and chest X-ray were more frequently prescribed to children admitted to the intensive care unit (antibiotics: OR 4.2, 95%CI 1.3–13.5; chest X-ray: OR 19.4, 95%CI 7.4–50.6). Latest prescription rates were all below the achievable benchmarks of care. In summary, following the latest American Academy of Pediatrics guideline, older, atopic children with wheezing and infants admitted to the intensive care unit were more likely to receive nonevidence-based interventions during an episode of bronchiolitis. These patient profiles are generally excluded from bronchiolitis trials, and therefore not specifically covered by the current guideline. Further research should focus on the benefit of bronchiolitis interventions in these particular populations.

from https://doi.org/10.26037/yareta:
pxcrz2cuhrhlngrip52357m2du.

**Funding:** This study was supported by the
Schmidheiny Foundation (F.S., no grant number,
http://www.fondation-schmidheiny.ch/lafondation.
html) and by the University of Geneva's Research
and Development Projects Fund (F.S., grant
number PRD 5-2017-I, https://www.hug.ch/
direction-medicale-qualite/recherche-clinique). The
funders had no role in study design, data collection
and analysis, decision to publish, or preparation of
the manuscript. There was no additional external
funding received for this study.

**Competing interests:** The authors have declared
that no competing interests exist.

## Introduction

Acute bronchiolitis is among the most common illnesses in pediatrics, both in the ambulatory
and in the hospital setting, and is associated with steadily rising annual costs [1, 2]. Whereas
the diagnosis of bronchiolitis should be based on clinical presentation, management should
essentially focus on symptoms, with minimal handling, and with hydration and oxygen when
required [3–7]. Furthermore, in order to limit the use of non-evidence-based therapies and
investigations, the 2014 American Academy of Pediatrics (AAP) clinical practice guideline on
bronchiolitis provided explicit recommendations against routine use of bronchodilators, corti-
costeroids, antibiotics, and chest radiography [3].

Avoiding unnecessary interventions not only benefits the patient, it is also essential to
counteract rising healthcare costs. Many studies have reported increasing adherence to the
2014 AAP guideline, with decreasing use of nonevidence-based interventions, mainly bron-
chodilators [8–11]. Achievable benchmarks of care (ABC) for bronchiolitis have also been
defined [12, 13] in order to improve the quality of healthcare processes and thereby reduce the
number of unnecessary interventions. The guideline acknowledges heterogeneity in patient
characteristics and in acute clinical presentations, however, it delegates prescription decisions
to the treating physician where there is a lack of guidance for specific patient profiles. Interven-
tions may, therefore, be beneficial in a selection of patients not covered by the guidelines [14–
16], and furthermore there is currently no detailed information regarding the characteristics
of these subjects still receiving these interventions.

In patients with acute bronchiolitis whose management was assessed and compared with
current achievable benchmarks of care, we aimed to identify profiles associated with the pre-
scription of bronchiolitis interventions. As these interventions are not recommended in the
2014 AAP guideline, we aimed to explore the rationale behind these prescriptions and evaluate
patient groups that should be included in a future revised guideline. The objective was hence
to evaluate whether we had moved beyond the "are we prescribing less", toward "our prescrip-
tions are well targeted".

## Methods

### Study design and subjects

In this single-centre retrospective observational study, the management of bronchiolitis pre-
guideline (Period 1: 2010 to 2012) was compared with two periods post-2014 AAP guideline
(Period 2: 2015 to 2016, early post-guideline; and Period 3: 2017 to 2018, late post-guideline).
All infants <1 year of age who presented for acute bronchiolitis at the emergency department
(ED) of the University Hospitals of Geneva during these periods were eligible. Acute bronchi-
olitis episodes were identified base on the International Classification of Diseases (ICD), ver-
sion 2010, using the code "acute bronchiolitis (J21)" and its derivatives. Medical records were
reviewed, and only patients with a typical clinical presentation were included [3]. This was
defined as a history and clinical exam compatible with bronchiolitis, namely a combination of
respiratory distress (e.g. tachypnoea, grunting, nasal flaring, retractions, and/or need for sup-
plementary oxygen) with fine crackles on auscultation. We excluded episodes that were mis-
coded, as well as individuals with chronic diseases, such as underlying pulmonary diseases
(e.g. bronchopulmonary dysplasia, cystic fibrosis, and congenital malformation), congenital
heart diseases, genetic diseases (e.g. Down syndrome), congenital or acquired immunodefi-
ciencies, or neuromuscular disorders, as well as those with more than 3 previous episodes of
bronchiolitis. Premature infants without chronic diseases were not excluded. The study was
approved by the Human Research Ethics Committee of the Canton of Geneva, who waived the

need for individual consent; the data was anonymized before analysis. This report followed the STROBE reporting guideline for cohort studies.

## Data extraction

The following data were collected from the medical charts: demographics (age, sex, and gestational age at birth), underlying medical conditions, date of admission, radiological assessment (chest X-ray), medications received (inhaled bronchodilator (β-agonist), oral corticosteroids, inhaled corticosteroids and antibiotics) before presentation to the ED (approximate reflection of private practice prescription) and in hospital, as well as disease severity parameters, such as need for hospitalization, length of stay (LOS; defined as the time interval between ED triage and hospital discharge order), admission to intensive care unit (ICU), requirement of nasogastric tube and of supplemental oxygen. Clinical parameters at ED presentation were also collected, including oxygen saturation, respiratory rate, presence of wheezing, and degree of retractions. The latter was divided for this analysis into "none/mild", defined as no retraction or retraction in only one site (e.g., subcostal retraction or nasal flaring), and "moderate/severe", defined as retractions in more than on site (e.g., supraclavicular and sternal retractions).

## Setting

The University Hospitals of Geneva includes one of the largest paediatric tertiary hospitals in Switzerland, with over 30,000 ED encounters per year. The institutional protocol on bronchiolitis management was changed following the publication of the 2014 AAP guideline [3]. Pre 2014 AAP guideline publication (period 1), a trial of bronchodilator treatment was recommended at ED presentation, especially in patients older than 6 months, but maintained only if beneficial (i.e. increased oxygen saturation, and decreased respiratory rate and wheezing). After 2014 (periods 2 and 3), supportive care was strongly recommended at ED admission and all other treatments (i.e. bronchodilators and corticosteroids) were discouraged [3]. At neither of the three periods were oral steroids routinely used, and antibiotics were only administered when a concomitant bacterial infection (e.g., pneumonia, acute otitis media) was clinically suspected. Laboratory testing and chest X-ray were at the discretion of the pediatrician in charge, as was the decision to start empiric antibiotic treatment; Laboratory confirmation of bacterial infection was not required before starting antibiotics. Hospitalization criteria were: oral intake less than 50% of daily nutritional requirements and requiring nasogastric tube feeding; low oxygen saturation requiring supplemental oxygen (less than 92% in period 1, and less than 90% in periods 2 and 3); progressive respiratory failure, apnoea, bradycardia, or poor social conditions requiring close monitoring. All patients requiring hospitalization for a respiratory illness during the RSV or influenza season had a viral diagnostic test to enable room cohorting. Room cohorting meant that we were not able to include viral testing in our analyses as it would not have been representative.

## Statistical analysis

All analyses were 2-tailed and performed using Stata statistical software version 16.0 (StataCorp). For subjects with more than one ED presentation for acute bronchiolitis during the period of interest, only the first episode was included in the analysis. First, subject and episode characteristics were compared between periods 1 and 2, and 1 and 3 using Kruskall-Wallis test for continuous variables, and $\chi 2$ or Fisher tests were used for categorical variables, as appropriate. Second, the proportion of subjects receiving medications before ED presentation or in hospital, as well as radiological assessments in hospital were reported with 95% confidence

intervals (CI) for each period; results from period 1 (pre-guideline) were compared to period 2 (early post-guideline) and to period 3 (late post-guideline), respectively, and reported as proportion difference with 95% CI. All results were then compared to the published ABC [12, 13]. Third, characteristics of subjects receiving non-recommended interventions (bronchodilators, oral corticosteroids, inhaled corticosteroids, antibiotics and chest X-ray) in hospital after the publication of the AAP guideline (periods 2 and 3) were compared to those who did not receive them using Kruskall-Wallis test for continuous variables, and χ2 or Fisher tests for categorical variables, as appropriate. Factors associated with use of these interventions were identified using univariate logistic regressions and reported as odds ratio (OR). Significant factors (p-value < 0.2) were included as possible covariates in multivariate models for each of the interventions. The models were created using backward stepwise exclusion of factors with p-values > 0.05. Given the slightly higher proportion of severe episodes in periods 2 and 3, sensitivity analyses were done, restricted to patients with severe retraction.

## Results

### Subject characteristics

A total of 963 subjects were included (434 in period 1, 216 in period 2, and 313 in period 3); patient characteristics are presented in Table 1. There was a significant lower proportion of females in period 1, and a higher proportion of severe episodes in periods 2 and 3, with longer duration of symptoms prior to ED presentation, as well as higher rates of moderate or severe retraction, of nasogastric tube requirement, and of hospitalization.

**Table 1. Characteristics of patients presenting with acute bronchiolitis episodes, by study period and in relation to the American Academy of Pediatrics clinical practice guideline on bronchiolitis [3].**

| Patient characteristics | Period 1 (2010–2012) | | Period 2 (2015–2016) | | Period 3 (2017–2018) | | p-value |
|---|---|---|---|---|---|---|---|
| | n | | n | | n | | |
| Age, months | 434 | 5.5 (2.7 to 7.8) | 216 | 5.0 (2.0 to 8.0) | 313 | 5.2 (2.6 to 7.7) | 0.6 |
| Sex, female | 434 | 145 (33.4) | 216 | 92 (42.6) | 313 | 138 (44.1) | 0.006 |
| Prematurity | 434 | 28 (6.5) | 206 | 24 (11.7) | 285 | 25 (8.8) | 0.08 |
| Personal history of atopy | 434 | 15 (3.5) | 216 | 13 (6.0) | 313 | 14 (4.5) | 0.3 |
| Family history of atopy | 155 | 76 (49.0) | 30 | 21 (70.0) | 47 | 18 (38.3) | 0.03 |
| Duration of symptoms at presentation, days | 431 | 2 (1 to 3) | 213 | 3 (2 to 4) | 297 | 3 (2 to 4) | <0.001 |
| Moderate or severe retractions | 434 | 168 (38.7) | 216 | 127 (58.8) | 313 | 195 (62.3) | <0.001 |
| Wheezing | 434 | 193 (44.5) | 216 | 103 (47.7) | 313 | 128 (40.9) | 0.3 |
| Tachypnoea | 432 | 276 (63.9) | 214 | 125 (58.4) | 310 | 180 (58.1) | 0.2 |
| Oxygen saturation at presentation, % | 425 | 97 (95 to 99) | 209 | 97 (95 to 99) | 302 | 97 (96 to 99) | 0.3 |
| Hospitalization | 434 | 128 (29.5) | 216 | 82 (38.0) | 313 | 112 (35.8) | 0.06 |
| ICU admission | 434 | 13 (3.0) | 216 | 15 (6.9) | 313 | 14 (4.5) | 0.07 |
| Length of stay in hospitalized subjects, days | 128 | 4.1 (2.9 to 5.9) | 82 | 4.2 (2.2 to 7.2) | 112 | 4.6 (2.8 to 6.8) | 0.9 |
| Required oxygen supplementation | 434 | 107 (24.7) | 216 | 45 (20.8) | 313 | 70 (22.4) | 0.5 |
| Required nasogastric tube | 434 | 65 (15.0) | 216 | 50 (23.1) | 313 | 65 (22.4) | 0.02 |
| Otitis media | 434 | 71 (16.4) | 216 | 33 (15.3) | 313 | 49 (15.7) | 0.9 |
| Pneumonia | 434 | 5 (1.2) | 216 | 3 (1.4) | 313 | 2 (0.6) | 0.7 |

Categorical variables are reported as number (%), continuous variables are reported as median (interquartile range).

AAP, American Academy of Pediatrics.

## Concordance of prescriptions with benchmark criteria

During period 1, only bronchodilators were prescribed above ABC (Table 2 and Fig 1). In periods 2 and 3, a sustained decrease in the proportion of subjects receiving bronchodilators was observed both before ED presentation and in hospital. In period 3, prescriptions for bronchodilators, antibiotics, oral corticosteroids, and chest X-ray all concorded with the highest ABC (Table 2 and Fig 1). In-hospital antibiotic prescriptions did increase between periods 1 and 3 (+4.3%, 95%CI +0.6% to +8.1%), but remained below ABC (9% in period 3; ABC 17–19%). Sensitivity analyses restricted to patients with severe retraction showed similar results.

## Factors associated with bronchodilator use in hospital

Subjects receiving bronchodilators after the publication of the 2014 AAP guideline were more likely to be older than 6 months (OR 25.8, 95%CI 12.6 to 52.6), have at ED presentation a history of atopy (OR 3.5, 95%CI 1.5 to 7.5) and wheezing (OR 5.4, 95%CI 3.3 to 8.7; Table 3), and before ED presentation have been prescribed a bronchodilator (OR 4.2, 95%CI 2.6 to 6.85) and oral corticosteroids (OR 5.1, 95%CI 1.3 to 19.2). Subjects who did not receive a bronchodilator in hospital were more likely to be male, admitted to ICU and have longer hospital stays, although these results might all have been confounded by younger age at presentation, as they were no longer significant in the multivariate model that included only older age and wheezing. Indeed, most of the ICU admissions (98%) were before 6 months of age, and LOS were overall longer in children aged less than 6 months, with a median duration of 4.6 days (IQR 3.0 to 7.0) compared with 3.2 days (IQR 2.0 to 5.0, p<0.001) in older subjects.

In the comparison of periods following the 2014 AAP guideline, we observed an overall decrease in bronchodilator use, as well as modifications in characteristics of patients receiving bronchodilators (Table 4). Subjects receiving bronchodilators during period 3 were older (96%

**Table 2. Evolution of prescriptions against benchmark reference values for acute bronchiolitis, over three periods of time.**

|  | Prescribed diagnostic tests and treatments | Benchmarks, % | | Study data, n (%) | | | Difference with period 1 (pre-guideline) % (95% CI) | |
|---|---|---|---|---|---|---|---|---|
|  |  | PHIS database[a, 12] | Systematic review [13] | Period 1[b] | Period 2[b] | Period 3[b] | Period 2 (early post-guideline) | Period 3 (late post-guideline) |
|  |  |  |  | (2010–2012) | (2015–2016) | (2017–2018) |  |  |
|  |  | n = 14 882 | n = 12 114 | n = 434 | n = 216 | n = 313 |  |  |
| **Before ED presentation** | Bronchodilators | 19%[a] | 16% | 103 (23.7) | 35 (16.2) | 54 (17.3) | -7.5 (-1.2 to -13.9) | -6.5 (-0.7 to -12.3) |
|  | Oral corticosteroids | 6% [a] | 1% | 6 (1.4) | 2 (0.9) | 7 (2.2) | -0.5 (-2.1 to +1.2) | +0.9 (-1.1 to +2.8) |
|  | Inhaled corticosteroids | - | - | 2 (0.5) | 1 (0.5) | 6 (1.9) | 0 (-1.1 to +1.1) | +1.5 (-0.2 to +3.1) |
|  | Antibiotics | 19% [a] | 17% | 25 (5.8) | 19 (8.8) | 21 (6.7) | +3.0 (-1.3 to +7.4) | +0.9 (-2.6 to +4.5) |
| **In hospital** | Bronchodilators | 19% | 16% | 141 (32.5) | 55 (25.5) | 53 (16.9) | -7.0 (-14.3 to +0.3) | -15.6 (-21.6 to -9.5) |
|  | Oral corticosteroids | 6% | 1% | 5 (1.2) | 7 (3.2) | 7 (2.2) | +2.1 (-0.5 to +4.7) | +1.1 (-0.8 to +3.0) |
|  | Inhaled corticosteroids | - | - | 0 (0.0) | 1 (0.5) | 1 (0.3) | +0.5 (-0.4 to +1.4) | +0.3 (-0.3 to +0.9) |
|  | Antibiotics | 19% | 17% | 20 (4.6) | 18 (8.3) | 28 (8.9) | +3.7 (-0.5 to +7.9) | +4.3 (+0.6 to +8.1) |
|  | Chest radiography | 32% | 42% | 52 (12.0) | 19 (8.8) | 30 (9.6) | -3.2 (-8.1 to +1.7) | -2.4 (-6.9 to +2.1) |

CI: confidence interval; ED: emergency department; PHIS: Pediatric Health Information Systems.

[a] PHIS database includes tertiary care hospital data only, which are less suitable as a comparator for prescriptions made before ED presentation [12].

[b] The three periods of time were determined as follows: Period 1 preceded the publication of the American Academy of Pediatrics clinical practice guideline on bronchiolitis [3], whereas Period 2 immediately followed publication of the guideline and Period 3 covered year 3 post-guideline publication.

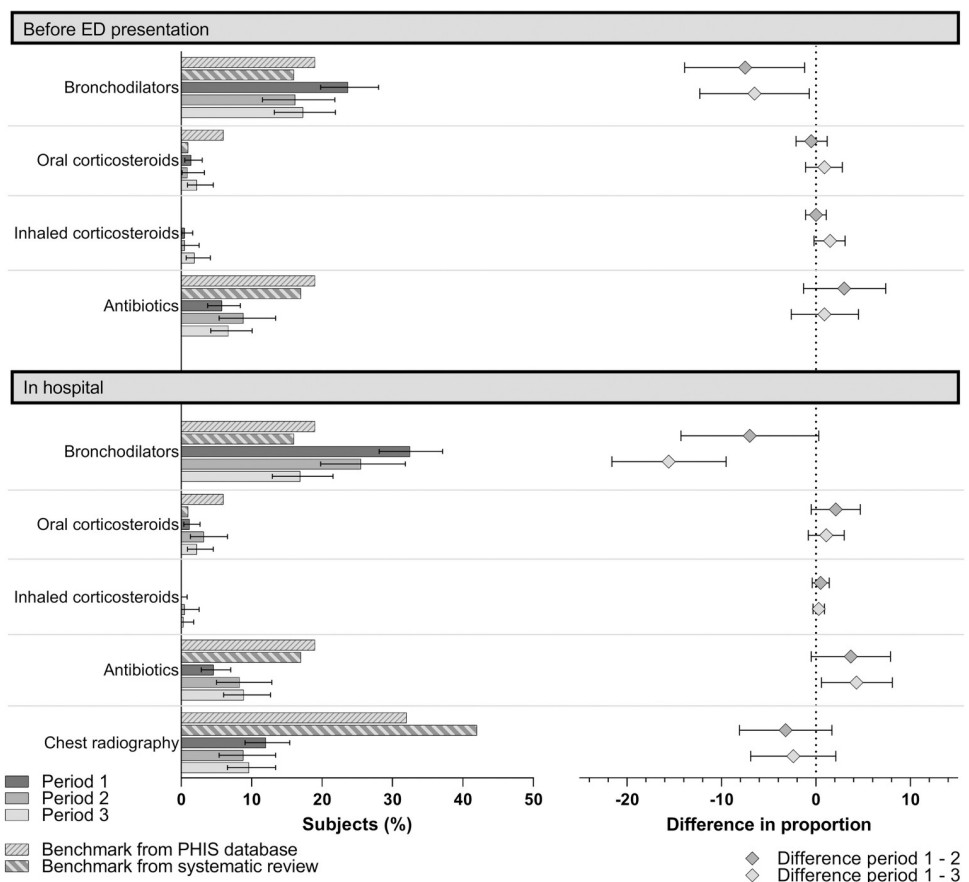

**Fig 1. Evolution of prescriptions compared to reference benchmarks in acute bronchiolitis over three periods of time and two clinical settings.** Period 1: 2010–2012, pre-AAP bronchiolitis guideline; period 2: 2015–2016, early post-guideline; period 3: 2017–2018, late post-guideline. Left panel: proportion of subjects, with 95% CI. Right panel: Differences in prescription rates between periods 1 and 2, and between periods 1 and 3, with 95% CI. Abbreviations: AAP, American Academy of Pediatrics; ED, emergency department; PHIS, Pediatric Health Information Systems.

were aged 6 months or more), had a more severe clinical presentation and longer duration of symptoms at ED presentation compared with those receiving bronchodilators in period 1.

### Factors associated with oral or inhaled corticosteroid use in hospital

Subjects receiving oral corticosteroids after the publication of the 2014 AAP guideline were more likely to be older than 6 months (OR 5.2, 95%CI 1.4 to 18.7) and present with wheezing (OR 4.9, 95% CI 1.3 to 17.8; Table 5). Only two subjects were still prescribed inhaled corticosteroids during period 2 and 3 (Table 6).

### Factors associated with antibiotic use in hospital

Subjects receiving antibiotics after publication of the 2014 AAP guideline were more likely to be older than 6 months (OR 2.0, 95%CI 1.1 to 3.8), have moderate or severe retraction (OR 3.5, 95%CI 1.5 to 7.5) and lower oxygen saturation (OR 0.8 per unit, 95%CI 0.8 to 0.9) at ED presentation, have been prescribed a bronchodilator (OR 2.4, 95%CI 1.2 to 4.7) and antibiotics (OR 6.6, 95%CI 3.1 to 14.1) before ED presentation, and were more likely to be hospitalized (OR 6.5, 95%CI 3.2 to 13.1) and admitted to ICU (OR 5.6, 95%CI 2.4 to 13.2; Table 7). In the

**Table 3. Characteristics of patients with acute bronchiolitis episodes and treated with or without bronchodilators in the emergency department after the publication of the American Academy of Pediatrics clinical practice guideline on bronchiolitis [3] (periods 2 and 3, combined).**

| Patient characteristics | No bronchodilator (N = 421) | | With bronchodilator (N = 108) | | p-value | Univariate | | Multivariate | |
|---|---|---|---|---|---|---|---|---|---|
| | n | | n | | | OR (95% CI) | p-value | OR (95% CI) | p-value |
| Age, % aged > 6 months | 421 | 126 (29.9) | 108 | 99 (91.7) | <0.001 | 25.8 (12.6 to 52.6) | <0.001 | 24.5 (11.8 to 50.8) | <0.001 |
| Age (among those > 6 months), months | 126 | 7.8 (6.9 to 9.1) | 99 | 9.0 (7.7 to 10.6) | <0.001 | 1.5 (1.2 to 1.7) | <0.001 | - | - |
| Sex, female | 421 | 194 (46.1) | 108 | 36 (33.3) | 0.02 | 0.6 (0.4 to 0.9) | 0.02 | - | - |
| Prematurity | 395 | 37 (9.4) | 96 | 12 (12.5) | 0.4 | 1.4 (0.7 to 2.8) | 0.4 | - | - |
| Personal history of atopy | 421 | 15 (3.6) | 108 | 12 (11.1) | 0.001 | 3.5 (1.5 to 7.5) | 0.003 | - | - |
| Family history of atopy | 45 | 23 (51.1) | 32 | 16 (50.0) | 0.9 | 0.9 (0.4 to 2.4) | 0.9 | - | - |
| Duration of symptoms at presentation, days | 406 | 3 (2 to 4) | 104 | 3 (2 to 4) | 0.06 | 0.9 (0.8 to 1.0) | 0.09 | - | - |
| Moderate or severe retractions | 421 | 251 (59.6) | 108 | 71 (65.7) | 0.2 | 1.3 (0.8 to 2.0) | 0.2 | - | - |
| Wheezing | 421 | 150 (35.6) | 108 | 81 (75.0) | <0.001 | 5.4 (3.3 to 8.7) | <0.001 | 5.0 (2.9 to 8.6) | <0.001 |
| Tachypnoea | 416 | 239 (57.4) | 108 | 66 (61.1) | 0.5 | 1.2 (0.7 to 1.8) | 0.5 | - | - |
| Oxygen saturation at presentation, % | 406 | 97 (96 to 99) | 105 | 97 (94 to 98) | 0.03 | 0.9 (0.9 to 1.0) | 0.08 | - | - |
| Hospitalization | 421 | 159 (37.8) | 108 | 35 (32.4) | 0.3 | 0.8 (0.5 to 1.2) | 0.3 | - | - |
| ICU admission | 421 | 28 (6.6) | 108 | 1 (0.9) | 0.02 | 0.1 (0.2 to 0.9) | 0.05 | - | - |
| Length of stay in hospitalized subjects, days | 159 | 4.6 (2.9 to 7.1) | 35 | 2.8 (1.6 to 5.1) | <0.001 | 0.7 (0.6 to 0.9) | 0.002 | - | - |
| Bronchodilators before ED presentation | 421 | 50 (11.9) | 108 | 39 (36.1) | <0.001 | 4.2 (2.6 to 6.85) | <0.001 | - | - |
| Oral corticosteroids before ED presentation | 421 | 4 (0.9) | 108 | 5 (4.6) | 0.008 | 5.1 (1.3 to 19.2) | 0.02 | - | - |
| Inhaled corticosteroids before ED presentation | 421 | 4 (0.9) | 108 | 3 (2.8) | 0.1 | 3.0 (0.7 to 13.5) | 0.2 | - | - |
| Antibiotics before ED presentation | 421 | 28 (6.6) | 108 | 12 (11.1) | 0.1 | 1.7 (0.9 to 3.6) | 0.1 | - | - |

Categorical variables are reported as number (%), continuous variables are reported as median (interquartile range).

**Table 4. Characteristics of patients with acute bronchiolitis episodes and treated with bronchodilators in the emergency department at study periods preceding and following publication of the American Academy of Pediatrics clinical practice guideline on bronchiolitis [3].**

| Patient characteristics | Period 1[a] (2010–2012) N = 141 | | Period 2[a] (2015–2016) N = 55 | | Period 3[a] (2017–2018) N = 53 | | p-value |
|---|---|---|---|---|---|---|---|
| | n | | n | | n | | |
| Age > 6 months | 141 | 93 (66) | 55 | 48 (87) | 53 | 51 (96) | <0.001 |
| Age if ≥ 6 months, months | 93 | 7.9 (7.1 to 9.3) | 48 | 8.5 (8.1 to 9.8) | 51 | 9.2 (7.6 to 10.8) | 0.1 |
| Sex, female | 141 | 42 (30) | 55 | 17 (31) | 53 | 19 (36) | 0.7 |
| Prematurity | 141 | 11 (8) | 51 | 5 (10) | 45 | 7 (16) | 0.3 |
| Personal history of atopy | 141 | 9 (6) | 55 | 8 (15) | 53 | 4 (8) | 0.2 |
| Familial history of atopy | 61 | 36 (59) | 15 | 9 (60) | 17 | 7 (41) | 0.4 |
| Duration of symptoms at presentation, days | 141 | 2 (1 to 3) | 54 | 3 (1 to 4) | 50 | 3 (2 to 4) | 0.01 |
| Moderate or severe retractions | 141 | 71 (50) | 55 | 38 (69) | 53 | 33 (63) | 0.04 |
| Wheezing | 141 | 107 (76) | 55 | 43 (78) | 53 | 38 (72) | 0.07 |
| Tachypnoea | 141 | 99 (70) | 55 | 32 (58) | 53 | 34 (64) | 0.3 |
| Oxygen saturation at presentation, % | 139 | 96 (94 to 98) | 53 | 96 (94 to 98) | 52 | 97 (96 to 98) | 0.2 |
| Hospitalization | 141 | 39 (28) | 55 | 18 (33) | 53 | 16 (30) | 0.8 |
| ICU admission | 141 | 2 (1) | 55 | 1 (2) | 53 | 0 (0) | 1 |
| Bronchodilators before ED presentation | 141 | 58 (41) | 55 | 20 (36) | 53 | 19 (36) | 0.7 |

Categorical variables are reported as number (%), continuous variables are reported as median (interquartile range)

[a] The three periods were determined so that Period 1 preceded the publication of the 2014 American Academy of Pediatrics clinical practice guideline on bronchiolitis [3], whereas Period 2 immediately followed publication of the guideline and Period 3 covered the 3rd year post-guideline publication.

**Table 5. Acute bronchiolitis episodes in patients treated with or without oral corticosteroids in the emergency department, after publication of the American Academy of Pediatrics clinical practice guideline on bronchiolitis [3] (periods 2 and 3, combined) [a].**

| Patient characteristics | No oral corticosteroids (N = 515) | | With oral corticosteroids (N = 14) | | p-value | Univariate | | Multivariate | |
|---|---|---|---|---|---|---|---|---|---|
| | n | | n | | | OR (95% CI) | p-value | OR (95% CI) | p-value |
| Age, % aged > 6 months | 515 | 214 (41.5) | 14 | 11 (78.6) | 0.006 | 5.2 (1.4 to 18.7) | 0.01 | 4.2 (1.1 to 15.2) | 0.03 |
| Age (among those > 6 months), months | 214 | 8.2 (7.2 to 9.9) | 11 | 8.6 (7.2 to 10.8) | 0.5 | 1.1 (0.8 to 1.6) | 0.5 | - | - |
| Sex, female | 515 | 225 (43.7) | 14 | 5 (35.7) | 0.6 | 0.7 (0.2 to 2.2) | 0.6 | - | - |
| Prematurity | 479 | 49 (10.2) | 12 | 0 (0) | 0.2 | - | - | - | - |
| Personal history of atopy | 515 | 25 (4.8) | 14 | 2 (14.2) | 0.1 | 3.2 (0.7 to 15.4) | 0.1 | - | - |
| Family history of atopy | 73 | 37 (50.7) | 4 | 2 (50.0) | 1.0 | 0.9 (0.12 to 7.3) | 1.0 | - | - |
| Duration of symptoms at presentation, days | 498 | 3 (2 to 4) | 12 | 2.5 (1.5 to 3) | 0.2 | 0.8 (0.6 to 1.1) | 0.3 | - | - |
| Moderate or severe retractions | 515 | 315 (61.2) | 14 | 7 (50.0) | 0.4 | 0.6 (0.2 to 1.8) | 0.4 | - | - |
| Wheezing | 515 | 220 (42.7) | 14 | 11 (78.6) | 0.008 | 4.9 (1.3 to 17.8) | 0.02 | 3.9 (1.1 to 14.4) | 0.04 |
| Tachypnoea | 510 | 298 (58.4) | 14 | 7 (50.0) | 0.5 | 0.7 (0.2 to 2.0) | 0.5 | - | - |
| Oxygen saturation at presentation, % | 497 | 97 (95 to 99) | 14 | 97.5 (94 to 100) | 0.7 | 0.9 (0.8 to 1.1) | 0.4 | - | - |
| Hospitalization | 515 | 189 (36.7) | 14 | 5 (35.7) | 0.9 | 0.9 (0.3 to 2.9) | 0.9 | - | - |
| ICU admission | 515 | 28 (5.4) | 14 | 1 (7.1) | 0.8 | 1.3 (0.2 to 10.6) | 0.8 | - | - |
| Length of stay in hospitalized subjects, days | 189 | 4.3 (2.6 to 6.9) | 5 | 4.1 (3.5 to 6.0) | 1.0 | 0.9 (0.7 to 1.3) | 1.0 | - | - |
| Bronchodilators before ED presentation | 515 | 84 (16.3) | 14 | 5 (35.7) | 0.06 | 2.8 (0.9 to 8.7) | 0.07 | - | - |
| Oral corticosteroids before ED presentation | 515 | 9 (1.7) | 14 | 0 (0) | 0.6 | - | - | - | - |
| Inhaled corticosteroids before ED presentation | 515 | 6 (1.2) | 14 | 1 (7.2) | 0.05 | 6.5 (0.7 to 58.1) | 0.09 | - | - |
| Antibiotics before ED presentation | 515 | 38 (7.4) | 14 | 2 (14.3) | 0.3 | 2.1 (0.4 to 9.7) | 0.3 | - | - |

Categorical variables are reported as number (%), continuous variables are reported as median (interquartile range).

[a] The study periods were determined based on the publication of the 2014 American Academy of Pediatrics clinical practice guideline on bronchiolitis: [3] Period 2 immediately followed publication of the guideline and Period 3 covered the 3[rd] year post-guideline publication.

multivariate model, the factors independently associated with in-hospital antibiotic use were older age (OR 17.3, 95%CI 5.9 to 51.2), ICU admission (OR 4.2, 95%CI 1.3 to 13.5), and length of stay (OR 1.2 per day, 95%CI 1.1 to 1.4; Table 7).

**Table 6. Acute bronchiolitis episodes in patients treated with or without inhaled corticosteroids in the emergency department, after publication of the American Academy of Pediatrics clinical practice guideline on bronchiolitis[3] (periods 2 and 3, combined)[a].**

| Patient characteristics | No inhaled corticosteroids (N = 527) | | With inhaled corticosteroids (N = 2) | | p-value | Univariate | | Multivariate | |
|---|---|---|---|---|---|---|---|---|---|
| | n | | n | | | OR (95% CI) | p-value | OR (95% CI) | p-value |
| Age, % aged > 6 months | 527 | 223 (42.3) | 2 | 2 (100.0) | 0.2 | - | - | - | - |
| Age (among those > 6 months), months | 223 | 8.2 (7.2 to 9.9) | 2 | 8.8 (6.8 to 10.8) | 1.0 | 1.1 (0.5 to 2.5) | 0.9 | - | - |
| Sex, female | 527 | 229 (43.4) | 2 | 1 (50.0) | 1.0 | 1.3 (0.1 to 20.9) | 0.9 | - | - |
| Prematurity | 490 | 49 (10.0) | 1 | 0 (0) | 1.0 | - | - | - | - |
| Personal history of atopy | 527 | 26 (4.9) | 2 | 1 (50.0) | 0.1 | 19.3 (1.2 to 316.8) | 0.04 | - | - |
| Family history of atopy | 76 | 38 (50.0) | 1 | 1 (100.0) | 1.0 | - | - | - | - |
| Duration of symptoms at presentation, days | 508 | 3 (2 to 4) | 2 | 2 (1 to 3) | 0.3 | 0.6 (0.3 to 1.6) | 0.4 | - | - |
| Moderate or severe retractions | 527 | 321 (60.9) | 2 | 1 (50.0) | 1.0 | 0.6 (0.04 to 10.3) | 0.8 | - | - |
| Wheezing | 527 | 229 (43.5) | 2 | 2 (100.0) | 0.2 | - | - | - | - |
| Tachypnoea | 522 | 305 (58.4) | 2 | 0 (0) | 0.2 | - | - | - | - |
| Oxygen saturation at presentation, % | 509 | 97 (95 to 99) | 2 | 100 (100 to 100) | 0.05 | - | - | - | - |
| Hospitalization | 527 | 193 (36.6) | 2 | 1 (50.0) | 1.0 | 1.7 (0.1 to 27.8) | 0.7 | - | - |
| ICU admission | 527 | 29 (5.5) | 2 | 0 (0) | 1.0 | - | - | - | - |
| Length of stay in hospitalized subjects, days | 193 | 4.3 (2.6 to 6.9) | 1 | 3.5 (3.5 to 3.5) | 0.6 | 0.8 (0.3 to 1.9) | 0.6 | - | - |
| Bronchodilators before ED presentation | 527 | 88 (16.7) | 2 | 1 (50.0) | 0.3 | 5.0 (0.3 to 80.5) | 0.3 | - | - |
| Oral corticosteroids before ED presentation | 527 | 9 (1.7) | 2 | 0 (0) | 1.0 | - | - | - | - |
| Inhaled corticosteroids before ED presentation | 527 | 6 (1.1) | 2 | 1 (50.0) | 0.03 | 86 (4.8 to 1556.6) | 0.002 | - | - |
| Antibiotics before ED presentation | 527 | 40 (7.6) | 2 | 0 (0) | 1.0 | - | - | - | - |

Categorical variables are reported as number (%), continuous variables are reported as median (interquartile range).

[a] The study periods were determined based on the publication of the American Academy of Pediatrics clinical practice guideline on bronchiolitis: [3] Period 2 immediately followed publication of the guideline and Period 3 covered year 3 post-guideline publication.

### Factors associated with chest X-ray prescription in hospital

Prescription of chest X-ray after the publication of the 2014 AAP guideline was associated with younger age (>6 months-old OR 0.3, 95%CI 0.2 to 0.7), moderate or severe retraction (OR 2.4, 95%CI 1.2 to 4.8) and lower oxygen saturation (OR 0.8 per unit, 95%CI 0.8 to 0.9) at ED presentation, hospitalization (OR 25.0, 95%CI 8.8 to 70.7), admission to ICU (OR 55.0, 95%CI 21.6 to 140.2) and longer hospital stay (OR 1.1 per day, 95%CI 1.0 to 1.2; Table 8). In the multivariate model, the factors independently associated with chest X-ray prescription were hospitalization (OR 13.4, 95%CI 4.6 to 39.5) and ICU admission (OR 19.4, 95%CI 7.4 to 50.6).

## Discussion

In this retrospective study of nearly 1000 episodes of bronchiolitis over 8 years, the management of acute bronchiolitis was evaluated in the light of the latest AAP guideline. We observed a sustained decrease in the use of bronchodilators not only in hospital, but also before ED presentation, reflecting changes in clinical practice at the hospital, but also among primary care physicians. However, despite new guideline recommendations, bronchodilators, and sometimes oral corticosteroids, continue to be prescribed in a minority of subjects over 6 months of age presenting with wheezing and an atopic background. Antibiotics and chest X-ray were more frequently prescribed to hospitalized infants admitted to ICU.

Heterogeneity in subject characteristics and in clinical presentation of acute bronchiolitis may be associated with variable responses to treatments [17–19]. Infants with an atopic

**Table 7. Acute bronchiolitis episodes in patients treated with or without antibiotics in the emergency department, after publication of the American Academy of Pediatrics clinical practice guideline on bronchiolitis [3] (periods 2 and 3, combined)[a].**

| Patient characteristics | No antibiotics (N = 483) | | With antibiotics (N = 46) | | p-value | Univariate | | Multivariate | |
|---|---|---|---|---|---|---|---|---|---|
| | n | | n | | | OR (95% CI) | p-value | OR (95% CI) | p-value |
| Age, % aged > 6 months | 483 | 198 (41.0) | 46 | 27 (58.7) | 0.02 | 2.0 (1.1 to 3.8) | 0.02 | 17.3 (5.9 to 51.2) | <0.001 |
| Age (among those > 6 months), months | 198 | 8.2 (7.3 to 9.9) | 27 | 9.4 (7.2 to 10.5) | 0.2 | 1.2 (0.9 to 1.5) | 0.1 | - | - |
| Sex, female | 483 | 212 (43.9) | 46 | 18 (39.1) | 0.5 | 0.8 (0.4 to 1.5) | 0.5 | - | - |
| Prematurity | 448 | 45 (10.0) | 43 | 4 (9.3) | 0.9 | 0.9 (0.3 to 2.7) | 0.9 | - | - |
| Personal history of atopy | 483 | 24 (4.9) | 46 | 3 (6.5) | 0.6 | 1.3 (0.4 to 4.6) | 0.6 | - | - |
| Family history of atopy | 66 | 34 (51.5) | 11 | 5 (45.4) | 0.7 | 0.8 (0.2 to 2.8) | 0.7 | - | - |
| Duration of symptoms at presentation, days | 466 | 3 (2 to 4) | 44 | 3 (2 to 5) | 0.2 | 1.1 (0.9 to 1.3) | 0.2 | - | - |
| Moderate or severe retractions | 483 | 287 (59.4) | 46 | 35 (76.1) | 0.03 | 2.2 (1.1 to 4.4) | 0.03 | - | - |
| Wheezing | 483 | 209 (43.3) | 46 | 22 (47.8) | 0.6 | 1.2 (0.6 to 2.2) | 0.6 | - | - |
| Tachypnoea | 478 | 273 (57.1) | 46 | 32 (69.6) | 0.1 | 1.7 (0.9 to 3.3) | 0.1 | - | - |
| Oxygen saturation at presentation, % | 468 | 97 (96 to 99) | 43 | 96 (93 to 97) | <0.001 | 0.8 (0.8 to 0.9) | <0.001 | - | - |
| Hospitalization | 483 | 159 (32.9) | 46 | 35 (76.1) | <0.001 | 6.5 (3.2 to 13.1) | <0.001 | - | - |
| ICU admission | 483 | 20 (4.1) | 46 | 9 (19.6) | <0.001 | 5.6 (2.4 to 13.2) | <0.001 | 4.2 (1.3 to 13.5) | 0.01 |
| Length of stay in hospitalized subjects, days | 159 | 4.3 (2.5 to 6.6) | 35 | 5.3 (3.2 to 7.8) | 0.1 | 1.1 (0.9 to 1.2) | 0.1 | 1.2 (1.1 to 1.4) | 0.003 |
| Bronchodilators before ED presentation | 483 | 75 (15.5) | 46 | 14 (30.4) | 0.01 | 2.4 (1.2 to 4.7) | 0.01 | - | - |
| Oral corticosteroids before ED presentation | 483 | 8 (1.7) | 46 | 1 (2.2) | 0.8 | 1.3 (0.2 to 10.8) | 0.8 | - | - |
| Inhaled corticosteroids before ED presentation | 483 | 5 (1.0) | 46 | 2 (4.3) | 0.06 | 4.3 (0.8 to 23.0) | 0.08 | - | - |
| Antibiotics before ED presentation | 483 | 27 (5.6) | 46 | 13 (28.3) | <0.001 | 6.6 (3.1 to 14.1) | <0.001 | - | - |

Categorical variables are reported as number (%), continuous variables are reported as median (interquartile range).
[a] The study periods were determined based on the publication of the American Academy of Pediatrics clinical practice guideline on bronchiolitis: [3] Period 2 immediately followed publication of the guideline and Period 3 covered year 3 post-guideline publication.

background, previous episodes of bronchiolitis, severe presentation or respiratory failure were excluded from most trials assessing bronchodilators in acute bronchiolitis, therefore the findings from those trials cannot be generalised to these children [20]. Older infants presenting with a wheezing phenotype will possibly benefit more from bronchodilator therapy [14–16]. We also observed a shift in age of the population receiving bronchodilators in period 3, in line

**Table 8. Acute bronchiolitis episodes in patients investigated with a chest X-ray in the emergency department, after publication of the American Academy of Pediatrics clinical practice guideline on bronchiolitis [3] (periods 2 and 3, combined)[a].**

| Patient characteristics | No chest X-ray (N = 480) | | With chest X-ray (N = 49) | | p-value | Univariate | | Multivariate | |
|---|---|---|---|---|---|---|---|---|---|
| | n | | n | | | OR (95% CI) | p-value | OR (95% CI) | p-value |
| Age, % aged > 6 months | 480 | 214 (44.6) | 49 | 11 (22.4) | 0.003 | 0.3 (0.2 to 0.7) | 0.004 | - | - |
| Age (among those > 6 months), months | 214 | 8.2 (7.2 to 9.8) | 11 | 9.9 (7.9 to 10.8) | 0.07 | 1.4 (0.9 to 2.0) | 0.07 | - | - |
| Sex, female | 480 | 206 (43.0) | 49 | 24 (49.0) | 0.4 | 1.3 (0.7 to 2.3) | 0.4 | - | - |
| Prematurity | 442 | 43 (9.7) | 49 | 6 (12.2) | 0.6 | 1.3 (0.5 to 3.2) | 0.6 | - | - |
| Personal history of atopy | 480 | 24 (5.0) | 49 | 3 (6.1) | 0.7 | 1.2 (0.4 to 4.3) | 0.7 | - | - |
| Family history of atopy | 69 | 35 (50.7) | 8 | 4 (50.0) | 1.0 | 1.0 (0.2 to 4.2) | 1.0 | - | - |
| Duration of symptoms at presentation, days | 462 | 3 (2 to 4) | 48 | 3 (2 to 4) | 0.8 | 0.9 (0.8 to 1.1) | 0.7 | - | - |
| Moderate or severe retractions | 480 | 284 (59.2) | 49 | 38 (77.5) | 0.01 | 2.4 (1.2 to 4.8) | 0.01 | - | - |
| Wheezing | 480 | 215 (44.8) | 49 | 16 (32.6) | 0.1 | 0.6 (0.3 to 1.1) | 0.1 | - | - |
| Tachypnoea | 475 | 273 (57.5) | 49 | 32 (65.3) | 0.2 | 1.4 (0.7 to 2.6) | 0.3 | - | - |
| Oxygen saturation at presentation, % | 469 | 97 (96 to 99) | 42 | 94 (93 to 97) | <0.001 | 0.8 (0.8 to 0.9) | <0.001 | - | - |
| Hospitalization | 480 | 149 (31.0) | 49 | 45 (91.8) | <0.001 | 25.0 (8.8 to 70.7) | <0.001 | 13.4 (4.6 to 39.5) | <0.001 |
| ICU admission | 480 | 7 (1.5) | 49 | 22 (44.9) | <0.001 | 55.0 (21.6 to 140.2) | <0.001 | 19.4 (7.4 to 50.6) | <0.001 |
| Length of stay in hospitalized subjects, days | 149 | 4.1 (2.3 to 6.0) | 45 | 6.0 (3.9 to 8.6) | 0.001 | 1.1 (1.0 to 1.2) | 0.007 | - | - |
| Bronchodilators before ED presentation | 480 | 83 (17.3) | 49 | 6 (12.2) | 0.4 | 0.7 (0.3 to 1.6) | 0.4 | - | - |
| Oral corticosteroids before ED presentation | 480 | 8 (1.7) | 49 | 1 (2.0) | 0.8 | 1.2 (0.1 to 10.0) | 0.8 | - | - |
| Inhaled corticosteroids before ED presentation | 480 | 6 (1.2) | 49 | 1 (2.0) | 0.6 | 1.6 (0.2 to 13.9) | 0.6 | - | - |
| Antibiotics before ED presentation | 480 | 38 (7.9) | 49 | 2 (4.1) | 0.3 | 0.5 (0.1 to 2.1) | 0.3 | - | - |

Categorical variables are reported as number (%), continuous variables are reported as median (interquartile range).

[a] The study periods were determined based on the publication of the American Academy of Pediatrics clinical practice guideline on bronchiolitis: [3] Period 2 immediately followed publication of the guideline and Period 3 covered year 3 post-guideline publication.

with our assumptions. These patients also presented with longer duration of symptoms and more severe presentation.

We also noticed increased antibiotic prescriptions over the years, in line with previous studies that also report that these rates are the hardest to improve [21–23]. However, the proportion of subjects receiving antibiotics in our study remained below benchmark values throughout all three periods. Antibiotics are sometimes justified in cases of pneumonia, or against co-infection, such as acute otitis media or urinary tract infections, which may be why ABC are set between 17% and 19% [12, 13]. Antibiotics are more likely to be prescribed in sicker children [21], although the proportion of severe episodes contributing to the rise in antibiotic prescription is unknown. In our setting, the rates of pneumonia remained stable, as did those of acute otitis media, despite a more stringent definition of the latter following the 2013 AAP guideline on acute otitis media [24]. However, in line with the literature [21], antibiotics were more frequently prescribed to the sickest children admitted to ICU.

The rate of chest radiography remained unchanged across all three study periods and was far below ABC [12, 13]. Other studies found a reduction in X-ray prescriptions following the 2014 AAP guideline recommendations, however, all studies had higher baseline prescription rates [25–27]. According to the literature, chest radiography only rarely contributes to improved management of acute bronchiolitis, while it causes unnecessary expenditures and radiation, increased rates of incorrect diagnosis of bacterial pneumonia, and unnecessary use of antibiotics, and should therefore be kept to a minimum [3, 28–31]. In our setting, chest

radiography was predominantly prescribed to hospitalized children admitted to ICU, possibly to rule out other pathologies.

Our results are, overall, in line with previous reports [8–11]. The strength of our study lies first, in that we were able to retrieve robust data from patients presenting with acute bronchiolitis between three distinct periods of observation. Second, we identified how clinical and imaging data from different settings were associated with prescriptions. Indeed, our study was based on data retrieved from medical charts that included clinical parameters as well as prescriptions for medications and investigations both pre-ED presentation and in hospital. In contrast, most published reports have relied on in-hospital data only, sometimes pre-dating current management guidelines, retrieved from physician questionnaires or hospital billing databases lacking clinical information, focussing only on medication prescriptions and ignoring investigations [8–11].

Our study has potential limitations inherent to its retrospective design. Given the tertiary hospital setting, our sample might have been biased toward greater disease severity and higher prescription rates; part of this limitation was mitigated by concomitant collection of data on treatments administered before ED presentation, although the latter is a slightly biased reflection of the ambulatory setting. Moreover, analysis was based on treatment prescription, which does not infer efficacy; standardised clinical evaluations pre- and post-treatment were not documented, precluding any conclusion on benefit.

We only included children less than 1 year of age, whereas ABC for bronchiolitis are based on children aged 0 to 2 years. In 1- to 2-year-olds, different phenotypes of bronchiolitis exist, and bronchodilator and corticosteroid use may be more often required because of the difficulty to differentiate between bronchiolitis and wheezing with an atopic phenotype. Phenotype-guided bronchodilator therapy strategy has recently been reported as more cost-effective, leading to less hospital admission and lower total treatment costs [32]. Lower ABC for bronchiolitis management in children less than 1 year of age might therefore be more suitable comparators than those used in the present study. Moreover, the AAP guideline ABC we used as comparators mostly reflect intervention rates preceding 2014; updated estimations might therefore be slightly lower. Also, one of the ABC based all its rates on tertiary care hospital data [12], which are less suitable as a comparator for prescriptions made before ED presentation. Lastly, we have also included patients with more than one episode of bronchiolitis; although some research group recommend to focus bronchiolitis research on first episodes (i.e. only in individuals without prior wheeze episode), this is not a consensus and heterogeneity in the population exists [33, 34].

In conclusion, we found that current prescription habits in our centre were compliant to available ABC for the management of bronchiolitis. Even though recourse to some of the interventions was reduced according to guideline recommendations, in this study we could show that they are still commonly used in older and atopic subjects with a severe wheezing presentation, or in children requiring hospitalization. The management of acute bronchiolitis is well-suited for "smarter medicine" in that the evidence-based 2014 AAP guideline does not recommend any specific intervention but aims rather to prevent overtreatment given the lack of evidence of benefit. Reducing unnecessary prescriptions can positively impact on intervention side effects and also help curb healthcare costs. However, several barriers still need to be overcome, such as parents' pressure to do more, apprehension of relying only on clinical assessment only, without investigations [35]. Targeted quality improvement strategies aiming at better patient and healthcare management should include education programs both for clinicians and parents.

Further research should account for patient heterogeneity rather than considering bronchiolitis as a homogenous disease. Focus should be on groups for whom interventions might be

beneficial but for whom there is no current guideline recommendation, such as patients with older age, severe presentation, atopic background, or recurrent episodes. Moreover, the ABC would benefit from a more up-to-date AAP guideline, and separate recommendations should be made for 0–1 and 1-2-year-old populations, for reasons already discussed. Finally, de-implementation campaigns that aim at stopping practices that are not evidence-based should account more for patient heterogeneity, and future guidelines should avoid general recommendations that overlook particularities in disease presentation until more conclusive evidence is available for each patient population [20, 36].

## Acknowledgments

We would like to thank Aliki Buhayer (www.prismscientific.ch) for editorial support.

## Author Contributions

**Conceptualization:** Constance Barazzone-Argiroffo, Alain Gervaix, Klara M. Posfay-Barbe, Annick Galetto-Lacour, Fabiola Stollar.

**Data curation:** Laure F. Pittet.

**Formal analysis:** Laure F. Pittet, Alban Glangetas.

**Funding acquisition:** Fabiola Stollar.

**Investigation:** Alban Glangetas.

**Methodology:** Laure F. Pittet, Constance Barazzone-Argiroffo, Alain Gervaix, Klara M. Posfay-Barbe, Annick Galetto-Lacour, Fabiola Stollar.

**Project administration:** Fabiola Stollar.

**Resources:** Fabiola Stollar.

**Supervision:** Fabiola Stollar.

**Visualization:** Laure F. Pittet.

**Writing – original draft:** Laure F. Pittet.

**Writing – review & editing:** Constance Barazzone-Argiroffo, Alain Gervaix, Klara M. Posfay-Barbe, Annick Galetto-Lacour, Fabiola Stollar.

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
