## [Decision Letter · Decision Letter 0]

10 Apr 2023

PONE-D-22-27995Factors associated with nonadherence to the American Academy of Pediatrics 2014 bronchiolitis guidelines: a retrospective studyPLOS ONE

Dear Dr. Pittet,

Thank you for submitting your manuscript to PLOS ONE. After careful consideration, we feel that it has merit but does not fully meet PLOS ONE’s publication criteria as it currently stands. Therefore, we invite you to submit a revised version of the manuscript that addresses the points raised during the review process.

Make sure the reference style is adjusted to PLOS One. What was the method used to assess suspicion of bacterial infection? Prior to administering antibiotics, was laboratory confirmation of bacterial infection required?

We look forward to receiving your revised manuscript.

Kind regards,

Sebastien Kenmoe

Academic Editor

PLOS ONE

Journal Requirements:

“This study was supported by the Schmidheiny Foundation (F.S., no grant number, http://www.fondation-schmidheiny.ch/lafondation.html) and by the University of Geneva’s Research and Development Projects Fund (F.S., grant number PRD 5-2017-I, https://www.hug.ch/direction-medicale-qualite/recherche-clinique). The funders had no role in study design, data collection and analysis, decision to publish, or preparation of the manuscript.”

Reviewers' comments:

Reviewer's Responses to Questions

**Comments to the Author**

1. Is the manuscript technically sound, and do the data support the conclusions?

Reviewer #1: Yes

Reviewer #2: Partly

2. Has the statistical analysis been performed appropriately and rigorously? 

Reviewer #1: Yes

Reviewer #2: Yes

3. Have the authors made all data underlying the findings in their manuscript fully available?

Reviewer #1: Yes

Reviewer #2: Yes

4. Is the manuscript presented in an intelligible fashion and written in standard English?

Reviewer #1: Yes

Reviewer #2: Yes

5. Review Comments to the Author

Reviewer #1: Thank you for submitting this manuscript, which highlights the benefits of published management guidelines and standards of care, but also highlights the deficiencies with these - namely the difficulty in extrapolating general guidelines for specific patient presentations.

I just have a few questions regarding the methodology and analysis.

Could you please expand more on the clinical definition of bronchiolitis used? "Only those with a typical clinical presentation were included". Can you please advise what the parameters for inclusion and exclusion based on "typical presentation" were?

The "post-guidelines" group were sicker at presentation to ED than the earlier cohort. Is there anything to suggest that the later presentation was also related to the guidelines publication? Eg did general practitioners feel more confident managing sicker babies in the community, leading to later presentation in ED?

Your pre-guidelines cohort were already receiving less antibiotics and less CXRs than the Benchmarking. Is this unique to your centre or is this similar Nationally?

Reviewer #2: Dear authors,

This is a retrospective, single-center study to evaluate the factors associated with nonadherence to the American Academy of Pediatrics 2014 bronchiolitis guidelines.

While I appreciate the relevance of the study and your efforts, there are some comments that I want to make to improve the manuscrit and its understanding:

-The main problem of this study is the selected population. If we stick to the definition of bronchiolitis, this is only the first episode in patients under 2 years of age. In this study, on the other hand, patients with up to 3 episodes of respiratory distress, which do not correspond to bronchiolitis itself, are included, and this produces a bias in the results and their interpretation. The management of these patients is not included in the AAP guidelines since it corresponds to another concept.

This is probably why patients older than 6 months with wheezing use more bronchodilators and corticosteroids, because they are treated as bronchospasms as they are not first episodes and, therefore, not bronchiolitis.

Lines 247-250 should be changed. As explained, if there are previous episodes of bronchiolitis, they are no longer considered as bronchiolitis.

-The other big problem with this manuscript is the results. The groups have statistically significant differences in their characteristics (for example, in severity), so I don't know if the results or their interpretations are really comparable. For this reason, it would be important to analyze subgroups with equal severity to avoid bias.

- Why were patients with some risk factor (heart disease, bronchopulmonary dysplasia...) excluded? It would be very interesting to be able to see what happens with this a priori more serious part of the population.

-It would be very interesting to use severity scores to know which patients we are talking about and not just use "none/mild" and "moderate/severe" for retractions. In patients with acute bronchiolitis there are very good validated severity scores and thus also allow comparisons between groups of equal severity.

-I would like to know why the use of hypertonic saline or nebulized adrenaline, widely used treatments in these patients that appear in the guidelines, has not been analyzed, and instead inhaled corticosteroids, which are rarely used, are analyzed. In fact, the table of inhaled corticosteroids could be removed.

-Regarding the use of antibiotics, it would be interesting to show in the results the infection data of the patients (pneumonia, acute otitis media, urinary infection...). In the discussion section, reference is made to them but without providing the data and it is interesting to assess the increase in their prescription between periods 1 and 3.

-How do the authors explain why more antibiotics are prescribed to RSV-positive patients?

-The conclusions should be reduced a bit and put only those derived from this study. Talking about reducing costs, although it is true, is not discussed in this manuscript.

6. PLOS authors have the option to publish the peer review history of their article (what does this mean?). If published, this will include your full peer review and any attached files.

Reviewer #1: No

Reviewer #2: No

---

## [Author Response · Author response to Decision Letter 0]

21 Apr 2023

PONE-D-22-27995

Factors associated with nonadherence to the American Academy of Pediatrics 2014 bronchiolitis guidelines: a retrospective study

PLOS ONE

Dear Dr. Pittet,

Thank you for submitting your manuscript to PLOS ONE. After careful consideration, we feel that it has merit but does not fully meet PLOS ONE’s publication criteria as it currently stands. Therefore, we invite you to submit a revised version of the manuscript that addresses the points raised during the review process.

Make sure the reference style is adjusted to PLOS One. What was the method used to assess suspicion of bacterial infection? Prior to administering antibiotics, was laboratory confirmation of bacterial infection required?

Authors’ reply: 

Dear Editor, thank you for giving us the opportunity to submit a revised manuscript.

Ordering of laboratory testing (usually full blood count, inflammatory parameters such as CRP and procalcitonin, and sometimes hemocultures), urine sampling, and chest X-ray were at the discretion of the paediatrician in charge, as was the decision to start empiric antibiotic treatment. In our center, laboratory confirmation of bacterial infection is not required before starting antibiotics. We have added this precision in the manuscript (lines 106-109, page 5)

The reference style is now in complete accordance with PLOS One recommendation.

Journal Requirements:

Authors’ reply: We have verified that our manuscript follows the PLOS ONE formatting guidelines.

Authors’ reply: The need for individual consent was waived by the ethical committee, and the data was anonymized before analysis; this information has been added to the manuscript (page 4, line 81)

“This study was supported by the Schmidheiny Foundation (F.S., no grant number, http://www.fondation-schmidheiny.ch/lafondation.html) and by the University of Geneva’s Research and Development Projects Fund (F.S., grant number PRD 5-2017-I, https://www.hug.ch/direction-medicale-qualite/recherche-clinique). The funders had no role in study design, data collection and analysis, decision to publish, or preparation of the manuscript.”

Authors’ reply: The funding statement has been updated in the cover letter.

Authors’ reply: The dataset can now be found at the following DOI: 10.26037/yareta:pxcrz2cuhrhlngrip52357m2du

Reviewers' comments:

Reviewer's Responses to Questions

Comments to the Author

1. Is the manuscript technically sound, and do the data support the conclusions?

Reviewer #1: Yes

Reviewer #2: Partly

2. Has the statistical analysis been performed appropriately and rigorously? 

Reviewer #1: Yes

Reviewer #2: Yes

3. Have the authors made all data underlying the findings in their manuscript fully available?

Reviewer #1: Yes

Reviewer #2: Yes

4. Is the manuscript presented in an intelligible fashion and written in standard English?

Reviewer #1: Yes

Reviewer #2: Yes

5. Review Comments to the Author

Reviewer #1

Thank you for submitting this manuscript, which highlights the benefits of published management guidelines and standards of care, but also highlights the deficiencies with these - namely the difficulty in extrapolating general guidelines for specific patient presentations.

I just have a few questions regarding the methodology and analysis.

Could you please expand more on the clinical definition of bronchiolitis used? "Only those with a typical clinical presentation were included". Can you please advise what the parameters for inclusion and exclusion based on "typical presentation" were?

Authors’ reply: A ‘typical presentation’ was defined as a history and clinical exam compatible with bronchiolitis, namely a combination of respiratory distress (e.g. tachypnoea, grunting, nasal flaring, retractions, and/or need for supplementary oxygen) with fine crackles on auscultation. Episodes that were miscoded were excluded. We added this information to the manuscript (lines 72-75 page 4).

The "post-guidelines" group were sicker at presentation to ED than the earlier cohort. Is there anything to suggest that the later presentation was also related to the guidelines publication? Eg did general practitioners feel more confident managing sicker babies in the community, leading to later presentation in ED?

Authors’ reply: Indeed, this was our hypothesis, but we were unable to verify it. 

As requested by reviewer 2, we performed sensitivity analyses restricted to children with moderate/severe retraction at ED presentation and fond similar results.

Your pre-guidelines cohort were already receiving less antibiotics and less CXRs than the Benchmarking. Is this unique to your centre or is this similar Nationally?

Authors’ reply: Hartog et al. (reference 11 in the manuscript) performed a cross‐sectional online survey of all board‐certified pediatricians in Switzerland in November 2019. The reported use of therapies were compared with that reported in previous surveys done in 2001 and 2006. The authors found that there was an overall decrease in the prescription of therapeutics and interventions for acute bronchiolitis from 2001 to 2019. Overall 27% of respondents reported prescribing antibiotics “sometimes” or to “infants at risk”.

Reference: Hartog K et al. Acute bronchiolitis in Switzerland - Current management and comparison over the last two decades. Pediatr Pulmonol. 2022;57(3):734-43. (reference 11 in manuscript)

Reviewer #2: 

Dear authors,

This is a retrospective, single-center study to evaluate the factors associated with nonadherence to the American Academy of Pediatrics 2014 bronchiolitis guidelines.

While I appreciate the relevance of the study and your efforts, there are some comments that I want to make to improve the manuscrit and its understanding:

-The main problem of this study is the selected population. If we stick to the definition of bronchiolitis, this is only the first episode in patients under 2 years of age. In this study, on the other hand, patients with up to 3 episodes of respiratory distress, which do not correspond to bronchiolitis itself, are included, and this produces a bias in the results and their interpretation. The management of these patients is not included in the AAP guidelines since it corresponds to another concept.

This is probably why patients older than 6 months with wheezing use more bronchodilators and corticosteroids, because they are treated as bronchospasms as they are not first episodes and, therefore, not bronchiolitis.

Lines 247-250 should be changed. As explained, if there are previous episodes of bronchiolitis, they are no longer considered as bronchiolitis.

Authors’ reply: Thank you for your comment, we agree with you that researchers have often attempted to focus the population of children with bronchiolitis by limiting inclusion to infants < 1 year old with a first episode of bronchiolitis, but it is not a consensus and heterogeneity in the population may persist.1 Historically, there has been a lack of consistency in the definition of bronchiolitis, due in large part to the intrinsic underlying heterogeneity of the condition.2 We now discuss it in the limitations section of the discussion (lines 314-316, page 25).

References

1. Zorc JJ, Hall CB. Bronchiolitis: recent evidence on diagnosis and management. Pediatrics. 2010;125(2):342-9. (reference 34 in manuscript)

2. Dalziel SR, Haskell L, O'Brien S, Borland ML, Plint AC, Babl FE, et al. Bronchiolitis. Lancet. 2022;400(10349):392-406. (reference 33 in manuscript)

-The other big problem with this manuscript is the results. The groups have statistically significant differences in their characteristics (for example, in severity), so I don't know if the results or their interpretations are really comparable. For this reason, it would be important to analyze subgroups with equal severity to avoid bias.

Authors’ reply: We did a sensitivity analysis restricted to patients with moderate/severe retraction and had similar results. This is now reported in the methods (page 6, lines 134-135) and in the result section (page 9, lines 155-156).

- Why were patients with some risk factor (heart disease, bronchopulmonary dysplasia...) excluded? It would be very interesting to be able to see what happens with this a priori more serious part of the population.

Authors’ reply: We have excluded patients with chronic diseases because they have been excluded from the AAP guideline, and therefore the AAP guideline are not applicable to them.1 Moreover, patients with chronic diseases have been excluded from the calculation of the achievable benchmarks of care, precluding comparison if they had been included in our study.2

References:

1. Ralston SL, Lieberthal AS, Meissner HC, Alverson BK, Baley JE, Gadomski AM, et al. Clinical practice guideline: the diagnosis, management, and prevention of bronchiolitis. Pediatrics. 2014;134(5):e1474-502. (reference 3 in the manuscript)

2. Parikh K, Hall M, Mittal V, Montalbano A, Mussman GM, Morse RB, et al. Establishing benchmarks for the hospitalized care of children with asthma, bronchiolitis, and pneumonia. Pediatrics. 2014;134(3):555-62. (reference 12 in the manuscript)

-It would be very interesting to use severity scores to know which patients we are talking about and not just use "none/mild" and "moderate/severe" for retractions. In patients with acute bronchiolitis there are very good validated severity scores and thus also allow comparisons between groups of equal severity.

Authors’ reply: We agree with the reviewer, but given the retrospective setting of this study, we were unable to use scores due to missing information, and had to use what was uniformly available in the medical charts, namely the respiratory rate and the degree of retraction.

The "bronchiolitis severity scores" have been withdrawn from our hospital’s algorithm for the management of bronchiolitis, following the publication of systematic reviews showing that most clinical scores were informally developed and insufficiently validated.1,2

In a recent study, Rodriguez-Martinez et al. pointed out that there is an urgent need to develop better instruments and to validate them in a comprehensive way.3 

References:

1. Destino L, Weisgerber MC, Soung P, Bakalarski D, Yan K, Rehborg R, Wagner DR, Gorelick MH, Simpson P. Validity of respiratory scores in bronchiolitis. Hosp Pediatr. 2012 Oct;2(4):202-9.

2. Fernandes RM, Plint AC, Terwee CB, Sampaio C, Klassen TP, Offringa M, van der Lee JH. Validity of bronchiolitis outcome measures. Pediatrics. 2015 Jun;135(6):e1399-408.

3. Rodriguez-Martinez CE, Sossa-Briceño MP, Nino G. Systematic review of instruments aimed at evaluating the severity of bronchiolitis. Paediatr Respir Rev. 2018 Jan;25:43-57.

-I would like to know why the use of hypertonic saline or nebulized adrenaline, widely used treatments in these patients that appear in the guidelines, has not been analyzed, and instead inhaled corticosteroids, which are rarely used, are analyzed. In fact, the table of inhaled corticosteroids could be removed.

Authors’ reply: Our centre does not use hypertonic saline, and uses nebulized adrenaline only in selected very severe bronchiolitis, reason why they were not assessed in this study. If you think it is necessary and the editor also requires it, we agree to remove table 6 presenting the data on inhaled corticosteroids.

-Regarding the use of antibiotics, it would be interesting to show in the results the infection data of the patients (pneumonia, acute otitis media, urinary infection...). In the discussion section, reference is made to them but without providing the data and it is interesting to assess the increase in their prescription between periods 1 and 3.

Authors’ reply: Thank you for your comment, we agree with you, however, unfortunately, we are not able to include this information as we did not collect data on urinary tract infection or other bacterial infection. 

The number of otitis media and pneumonia are available in Table 1.

-How do the authors explain why more antibiotics are prescribed to RSV-positive patients?

Authors’ reply: All patients requiring hospitalization for a respiratory illness during the RSV or influenza season had a viral diagnostic test done to enable room cohorting. Therefore, viral testings (and positive test results) were only done in the sickest patient. As analyses were biased by this, the revised manuscript does no longer include any analysis on RSV and influenza test results. Thanks for pointing that out to us.

-The conclusions should be reduced a bit and put only those derived from this study. Talking about reducing costs, although it is true, is not discussed in this manuscript.

Authors’ reply: We like to put the results of our study in perspective, in order to emphasize why they are important. If you think it is necessary and the editor also requires it, we agree to remove the part of the discussion on cost (page 25, lines 305 to 307).

6. PLOS authors have the option to publish the peer review history of their article (what does this mean?). If published, this will include your full peer review and any attached files.

Do you want your identity to be public for this peer review? For information about this choice, including consent withdrawal, please see our Privacy Policy.

Reviewer #1: No

Reviewer #2: No

---

## [Decision Letter · Decision Letter 1]

27 Apr 2023

Factors associated with nonadherence to the American Academy of Pediatrics 2014 bronchiolitis guidelines: a retrospective study

PONE-D-22-27995R1

Dear Dr. Pittet,

We’re pleased to inform you that your manuscript has been judged scientifically suitable for publication and will be formally accepted for publication once it meets all outstanding technical requirements.

Kind regards,

Sebastien Kenmoe

Academic Editor

PLOS ONE

Additional Editor Comments (optional):

Reviewers' comments:

Reviewer's Responses to Questions

**Comments to the Author**

1. If the authors have adequately addressed your comments raised in a previous round of review and you feel that this manuscript is now acceptable for publication, you may indicate that here to bypass the “Comments to the Author” section, enter your conflict of interest statement in the “Confidential to Editor” section, and submit your "Accept" recommendation.

Reviewer #1: All comments have been addressed

Reviewer #2: All comments have been addressed

2. Is the manuscript technically sound, and do the data support the conclusions?

Reviewer #1: Yes

Reviewer #2: Yes

3. Has the statistical analysis been performed appropriately and rigorously? 

Reviewer #1: Yes

Reviewer #2: Yes

4. Have the authors made all data underlying the findings in their manuscript fully available?

Reviewer #1: Yes

Reviewer #2: Yes

5. Is the manuscript presented in an intelligible fashion and written in standard English?

Reviewer #1: Yes

Reviewer #2: Yes

6. Review Comments to the Author

Reviewer #1: (No Response)

Reviewer #2: I feel the author's have made necessary revisions or justified rationale for not including certain items.

7. PLOS authors have the option to publish the peer review history of their article (what does this mean?). If published, this will include your full peer review and any attached files.

Reviewer #1: No

Reviewer #2: No

---

## [Editor Report · Acceptance letter]

8 May 2023

PONE-D-22-27995R1 

Factors associated with nonadherence to the American Academy of Pediatrics 2014 bronchiolitis guidelines: a retrospective study 

Dear Dr. Pittet:

I'm pleased to inform you that your manuscript has been deemed suitable for publication in PLOS ONE. Congratulations! Your manuscript is now with our production department. 

Kind regards, 

on behalf of

Dr. Sebastien Kenmoe 

Academic Editor

PLOS ONE